# A retrospective analysis of prenatal genetic results in fetal hydronephrosis

**Keqin Jin**[1,2]*, **Xiayuan Xu**[1,2], **Yue Qian**[1,2], **Liping Zhang**[3], **Min Hu**[4,2]*, **Jianfeng Luo**[1,5]

**1** Genetic Laboratory, Jinhua Maternal & Child Health Care Hospital, Jinhua, Zhejiang, China, **2** Jinhua Key Laboratory for Comprehensive Prevention and Control of Birth Defects, Jinhua Maternal & Child Health Care Hospital, Jinhua, Zhejiang, China **3** Department of Ultrasound Medicine, Jinhua Maternal & Child Health Care Hospital, Jinhua, Zhejiang, China, **4** Gynaecology and Obstetrics, Jinhua Maternal & Child Health Care Hospital, Jinhua, Zhejiang, China, **5** Teaching Affairs Office, Affiliated Jinhua Hospital of Zhejiang University School of Medicine, Jinhua, Zhejiang, China

* jkq2239026@163.com (KJ); humin100@hotmail.com (MH)

## Abstract

### Objective

To discuss the application value of technologies such as chromosome microarray analysis (CMA) in fetuses with hydronephrosis and pyelectasis.

### Methods

Retrospectively collected the prenatal diagnostic data of 83 fetuses with hydronephrosis from January 2020 to July 2024. The positive rate of chromosomal abnormalities detected by different ultrasound abnormalities was statistically analyzed.

### Results

Among the 83 pregnant women, 10 cases of abnormal karyotypes were detected by invasive prenatal diagnosis, with an abnormality rate of 12.05%. Numerical chromosomal abnormalities accounted for 90%, mainly trisomy 21 and 13. In the fetuses with normal karyotype/no abnormality, CMA additionally detected 15 copy number variations (CNVs) in 12 cases. Divided into isolated hydronephrosis and non-isolated hydronephrosis groups, the detection rates of fetuses carrying pathogenic CNVs were 5.56% and 12.77% respectively, and the detection rates of fetuses carrying variants of uncertain significance (VUS) were 19.44% and 8.51% respectively. Still, the differences between the two groups were not statistically significant ($P > 0.05$). Divided into moderate to severe hydronephrosis group and mild hydronephrosis group, the detection rate of pathogenic abnormalities by CMA was 10% and 37.21% respectively, and the difference between the two groups was statistically significant ($P < 0.05$).

**Data availability statement:** The supporting data of this study are stored in the controlled - access data repository at National Genomics Data Center(https://ngdc.cncb.ac.cn/). OMIX ID: OMIX005796.Shared URL:https://ngdc.cncb.ac.cn/omix/release/OMIX005796 OMIX ID: OMIX008983.Shared URL:https://ngdc.cncb.ac.cn/omix/release/OMIX008983

**Funding:** This work was supported by the following: -Jinhua Science and Technology Bureau, (2019-3-002a), Mr Jianfeng Luo; -Jinhua Science and Technology Bureau, (2021-3-123), Mr Min Hu; -Jinhua Maternal & Child Health Care Hospital Research Incubation Fund General Project, (JHFB2023-2-10), Mr Keqin jin; -Jinhua Science and Technology Bureau, (2024-4-140), Mrs Yue Qian.

**Competing interests:** The authors declare no competing interests.

**Abbreviations:** CMA, chromosome microarray analysis; CNVs, copy number variations; VUS, variants of uncertain significance; CAKUT, Congenital abnormalities of the kidney and urinary tract; ESRD, end-stage renal disease; UPJO, ureteropelvic junction obstruction; UVJO, ureterovesical junction obstruction; VUR, vesicoureteric reflux; PUV, posterior urethral valves

## Conclusion

Hydronephrosis is associated with chromosomal abnormalities, and the rate of chromosomal abnormalities increases significantly as the degree of hydronephrosis increases. The combined use of CMA technology can detect abnormalities caused by chromosomal microdeletions and/or microduplications, which is of great value for clinical prenatal consultation.

---

Congenital abnormalities of the kidney and urinary tract (CAKUT) with hydronephrosis as a common manifestation are a series of diseases caused by abnormalities in the urinary collecting system, embryonic migration of the kidneys, or renal parenchymal development. CAKUT is one of the common fetal malformations in prenatal diagnosis, accounting for 15%-20% of all congenital system malformations [1,2], and poses a fatal threat to the fetus [3].Hydronephrosis is a physiological phenomenon in 64%-94% of cases, which can spontaneously resolve in late pregnancy or after birth; only 4.1%-15.4% are pathological [4], such as ureteropelvic junction obstruction (UPJO), ureterovesical junction obstruction (UVJO), vesicoureteric reflux (VUR), and posterior urethral valves (PUV), which will gradually worsen after birth and require surgical treatment.Studies by Smith J.M. et al. [5–7] have shown that CAKUT is the leading cause of end-stage renal disease (ESRD) in children and adolescents, and most ESRD patients in children require kidney transplantation or dialysis, with approximately 41.3% of patients needing kidney transplantation [8]. Early prenatal screening for fetal malformations is important for clinical prognosis assessment and intervention, which can help prevent and reduce the incidence of severe CAKUT.

Due to the normal development of the kidneys and ureters being regulated by various genes, molecules, and fetal environments, abnormalities in these factors can lead to obstruction of the ureteropelvic junction, resulting in hydronephrosis as the main manifestation of CAKUT (Congenital Anomalies of the Kidney and Urinary Tract). Rosenblum et al. [9] reported that in many factors, about 50% of CAKUT is caused by chromosomal abnormalities, copy number variations (CNVs), and single-gene genetic abnormalities that lead to changes in genetic material. Traditional karyotyping is a common prenatal diagnostic method, but most of the CNVs (<10Mb) are difficult to detect. Chromosome microarray analysis (CMA) has a resolution that is a thousand times higher than traditional karyotyping, which can validate NIPS (Noninvasive Prenatal Screening) results and detect chromosomal microdeletions/microduplications that cannot be detected by karyotyping. This retrospective study analyzed the prenatal diagnostic data of 83 pregnant women, aiming to explore the diagnostic value of CMA and other techniques for fetal hydronephrosis.

## Subjects and methods

### Ethics statement

The retrospective analysis of this study has been approved by the hospital's ethics committee (Ethics Review No.: 2024 - QT - 004). All procedures implemented within

this study were strictly in accordance with the ethical guidelines formulated by the ethics committee of Jinhua Maternal & Child Health Care Hospital. Written informed consent was duly obtained from every study participant. Moreover, all methods were carried out in full compliance with relevant guidelines and regulations, ensuring the integrity and ethical soundness of the entire research process.

## Information

This retrospective study collected data from fetuses with hydronephrosis who visited the Prenatal Diagnosis Center of Jinhua Maternal and Child Health Hospital from January 2020 to July 2024. Among them, 83 cases voluntarily underwent invasive prenatal diagnosis, with maternal ages ranging from 19 to 43 years old. All participants signed the informed consent form for the relevant tests.The inclusion criteria referred to the standards in the literature [10,11]. 40 cases were classified as mild hydronephrosis, with a renal pelvis anteroposterior diameter of 4-<7mm in the mid-trimester (16–27 weeks) or 7-<9mm in the late trimester (≥28 weeks). 43 cases were classified as moderate to severe hydronephrosis, with a renal pelvis anteroposterior diameter ≥7mm in the mid-trimester or ≥9mm in the late trimester, or other types of hydronephrosis.They were further divided into two groups based on whether other ultrasound abnormalities were present. The isolated group included 36 cases with only isolated hydronephrosis, while the non-isolated group included 47 cases with hydronephrosis combined with other malformations or soft markers.

## Ultrasonography

During routine prenatal ultrasound examinations of pregnant women, a GE Voluson 730 and/or GE Voluson E8 ultrasound Doppler machine (probe frequency 3.5 MHz) is used. The examination is performed in the supine position, using systematic ultrasound standard views for the pregnant woman.For the examination of the fetal kidneys, transverse, longitudinal, and oblique scans are utilized to assess the kidneys. The size of the kidneys is measured, and a detailed examination of the shape, position, structure, and internal echo characteristics is conducted. Coronal or longitudinal scanning is performed to observe the ureteral separation, dilation, and the size and filling state of the bladder.

## Karyotype analysis

Under the guidance of ultrasound monitoring, we graciously collect 18–20 milliliters of amniotic fluid under a sterile environment. Subsequently, this preserved sample is inoculated onto BIO-AMFTM-2 culture medium, based on the operational manual of our esteemed genetic laboratory. The amniotic fluid is nurtured routinely for an appropriate duration of eight to ten days. Post cultivation, we proceed with collection, standard preparation, and G banding, all according to the recognised international system ISCN (2016) standard for describing karyotypes in human cellular genetics.

## Chromosomal microarray analysis (CMA)

Prior to February 2022, our testing for CMA was entrusted to Beijing Beikang Medical Examination Institute. As of February 2022, we've transitioned to utilizing localized examination services internally.In compliance with the standardized process delineated by CMA analysis, employing the American Affymetrix CytoScan platform, specifically the CytoScan 750K Array Kit and Reagent Kit Bundle (catalogue number: 901859), for the extraction of whole genome DNA from a 10 ml human amniotic fluid sample, followed by genomic DNA digestion, ligation, amplification, purification, fragmentation, signal labelling, microarray hybridization, washing, scanning, as well as comprehensive data processing and analysis utilizing ChAS v.3.0 software, we report copy number variations (CNVs) that meet or exceed both a threshold value of ≥500kb repeat and ≥200kb deletion on chromosomes.

### Recall and follow-up of pregnant women

Regular physical examinations, as well as phone calls, are provided to expectant mothers undergoing preimplantation diagnosis to monitor the fetus.

### Statistical processing

In keeping with the methodology outlined by Sun Zhenquan [12], our statistical analysis performed on enumeration data utilizes a rate (%) as a representation, and for intergroup comparisons, we employ a chi-squared test which is significant at P<0.05 to indicate a statistically significant difference. The appropriate formula for this chi-squared test is selected based on specific criteria, and the results of these calculations are meticulously verified using the VassarStats: Statistical Computation Web Site (https://vassarstats.net/index.html).

1) Calculate when n > 40 and all T are greater than 5: $\chi^2 = \frac{(ad-bc)^2 n}{(a+b)(c+d)(a+c)(b+d)}$;

2) Utilize computation for n >40 with at least one T < 5: $\chi^2 = \frac{(|ad-bc|-\frac{n}{2})^2 n}{(a+b)(c+d)(a+c)(b+d)}$;

3) Apply calculation for both n ≤ 40 or T < 1: $Pi = \frac{(a+b)!(c+d)!(a+c)!(b+d)!}{a!b!c!d!n!}$.

## Results

### Acceptance of invasively obtained, prenatal diagnostic karyotype analysis results

A total of 83 voluntary invasive prenatal specimens were analyzed for chromosomal karyotype. The analysis detected 10 cases of abnormal karyotypes, with a variation rate of 12.05%.Numerical chromosome abnormalities accounted for 90% of the cases, primarily consisting of trisomy 21 and trisomy 13. Structural chromosome abnormalities accounted for 10% of the cases. Detailed results are shown in Table 1.

### Acceptance of CMA results for interventional prenatal diagnosis

Among the fetal cases with normal karyotypes or no obvious abnormalities, the chromosomal microarray analysis (CMA) additionally detected 15 copy number variations (CNVs) in 12 cases with fetal hydronephrosis.(Results are detailed in Table 2).

### Correlation analysis between renal hydronephrosis and fetal chromosome anomalies in both groups

In 83 CMA testing specimens, divided into two groups based on the presence of other ultrasound abnormalities, the detection rate of fetuses carrying pathogenic CNVs in the isolated hydronephrosis group was 5.56%, while in the non-isolated hydronephrosis group it was 12.77%. The detection rate of fetuses carrying VUS variants was 19.44% in the isolated hydronephrosis group and 8.51% in the non-isolated hydronephrosis group. However, the differences between the two groups were not statistically significant (P>0.05).The results are displayed in Table 3.

### Comparison of fetal anomaly detection rates with CMA in moderate to severe hydronephrosis groups versus mild hydronephrosis groups

In 83 specimens, divided into two groups based on the severity of hydronephrosis, the detection rate of fetuses carrying variants by CMA in the moderate-to-severe hydronephrosis group was 10%, while in the mild hydronephrosis group it was 37.21%. The difference between the two groups was statistically significant (P<0.05), as depicted in Table 4.

## Discussion

Congenital hydronephrosis (antenatal hydronephrosis, ANH) has a prenatal incidence rate of 1% to 5% [13]. Xu Hong et al. [11] reported that 21% to 28% of children with antenatal congenital hydronephrosis were normal on the first postnatal

**Table 1. Abnormal karyotypes and CMA results in nine fetuses with hydronephrosis.**

| Karyotype | Number of cases | Rate | CMA | Ultrasonography | Fetal Outcomes |
|---|---|---|---|---|---|
| Numerical Abnormalities | 8 | 90.0% | | | |
| 47,XN,+21 | 4 | 40.0% | arr (21)x3 | 1 case: hydronephrosis, NT 3.4 mm, nasal bone hypoplasia;1 case: hydronephrosis, NT 2.9 mm;1 case: hydronephrosis, nasal bone absence, ventricular echogenic spots1 case: hydronephrosis | Induced labor |
| 47,XN,+13 | 2 | 20.0% | arr (13)x3 | 1 case: hydronephrosis, holoprosencephaly, omphalocele and facial/cardiac anomalies; 1 case: hydronephrosis, mega bladder, hyper echoic intestines, and ventricular echogenic spots. | Induced labor |
| 47,XY,+18[4]/46,XY[46] | 1 | 10.0% | arr(1–22)×2,(XN)×1 | hydronephrosis | Induced labor |
| 47,XX,+15[2]/46,XX[108] | 1 | 10.0% | arr(15)x2-3 | hydronephrosis、Lone Right Kidney Dysplasia | Induced labor |
| 45,X[45]/46,XY[144] | 1 | 10.0% | arr(1–22)×2,(XN)×1 | hydronephrosis | Induced labor |
| Structural Variations | 1 | 10.0% | | | |
| 46,XY,?2 | 1 | 10.0% | arr[hg19] 2q12.2q14.2(106,873,993–120,408,323)x1 | hydronephrosis | Induced labor |

examination, among which 45% showed abnormalities in subsequent follow-ups. Therefore, it is necessary to screen for pathogenic factors prenatally to provide an important basis for clinical judgment and intervention. In this study, 83 cases of fetal hydronephrosis were screened by ultrasound for interventional prenatal diagnosis. Chromosomal karyotype analysis detected 10 cases of abnormal karyotypes, with a detection rate of 12.05%. CMA analysis detected 20 fetuses carrying 23 variants, with a variant detection rate of 24.1%, slightly higher than the 16.6% reported by Cai, M [14]. This indicates a close correlation between hydronephrosis abnormalities and chromosomal numerical abnormalities such as trisomy 21. CMA has higher resolution and a higher variant detection rate compared to karyotyping.. It can be seen that hydronephrosis abnormalities are closely related to chromosomal number abnormalities such as trisomy 21 syndrome. Furthermore, CMA improves resolution and increases the detection rate of variations compared to karyotyping.karyotyping, thereby augmenting variant detection.

Fetal microdeletion/microduplication syndromes caused by CNVs, although relatively rare, are diverse and have complex phenotypes, with an incidence of 1‰-3‰ in nature. Most CNVs result from rearrangements, which disrupt the balance between genes and may alter genetic material, leading to fetal developmental abnormalities [15]. In recent years, these are associated with congenital malformations. In this study, in addition to 7 cases of chromosomal number abnormalities detected by CMA, 16 CNVs were identified in 13 fetuses. One case involved a deletion of 13.5 Mb in the 2q12.2q14.2 region, including a deletion in the 2q13 recurrent region (includes BCL2L11) (chr2:111392193–113104742). The carrier exhibited a variety of symptoms, including developmental delay/intellectual disability, nonspecific malformations, abnormal head size (macrocephaly or microcephaly), congenital heart disease, and other nonspecific phenotypes. Among them, *NPHP1 (607100)* was also associated with autosomal recessive Senior-Loken syndrome type 1 (Phenotype MIM number: 266900). Additionally, 15 CNVs were identified in 12 fetuses with normal or variant chromosomal karyotypes but with renal pelvis dilatation. These CNVs involved chromosomes 1, 2, 3, 4, 6, 7, 11, 14, 22, and Y, but all were in regions of uncertain clinical significance.A coincidence may merit our attention, among the 12 fetuses with normal karyotypes, 3 cases involved deletions of the 7q11.21 segment. Zhang H et al. [16] previously reported that in 18 patients with 7q11.21 deletions, 2 prenatal fetuses also exhibited renal developmental abnormalities, suggesting a possible association between 7q11.21 deletion and renal developmental abnormalities. However, this region only includes one protein-coding

**Table 2. CMA Analysis of 10 Cases demonstrating Normal Variation or Absence of Abnormal Chromosomal Karyotypes.**

| No. | Ultrasonography | CMA | Banding | Size (Kb) | Pathogenicity classification | Parent Verification | Fetal Outcomes |
|---|---|---|---|---|---|---|---|
| 1 | Mild hydronephrosis、Ventricular septal defect | arr[hg19] 1q21.1(145,390,101–145,966,117)x3 arr[hg19]3p14.2(59,731,510–61,354,694)x3 | 1q21.1 | 576 | VUS | Mother: arr[hg19] 1q21.1(145,390,101–145,988,238)x3; Father: arr[hg19] 3p14.2(59,731,510–61,386,297)x3 | Continue Pregnancy |
| | | | 3p14.2 | 1,623 | VUS | | |
| 2 | hydronephrosis | arr[hg19] 2p22.3(34,002,380–35,045,602)x3 | 2p22.3 | 1,043 | VUS | | Continue Pregnancy |
| 3 | hydronephrosis | arr[hg19] 2q24.1(158,264,015–159,517,744)x1 | 2q24.1 | 1,253 | VUS | | Continue Pregnancy |
| 4 | Hydronephrosis, Permanent right umbilical vein | arr[hg19] 4q35.2(188,110,191–189,497,561)x3 | 4q35.2 | 1,387 | VUS | | Continue Pregnancy |
| 5 | hydronephrosis、NT3.2mm | arr[hg19] 6q24.2(143,516,141–144,063,784)x3 arr[hg19] 7q11.21(64,638,714–65,091,611)x1 | 6q24.2 | 547 | VUS | | Continue Pregnancy |
| | | | 7q11.21 | 453 | VUS | | |
| 6 | hydronephrosis | arr[hg19] 7q11.21(64,576,418–65,091,611)x1 | 7q11.21 | 515 | VUS | | Continue Pregnancy |
| 7 | hydronephrosis、ventricular echogenic spots、Posterior fossa cistern widened | arr[hg19] 7q11.21(64,612,880–65,162,169)x1 | 7q11.21 | 550 | VUS | | Continue Pregnancy |
| 8 | hydronephrosis | arr[hg19] 11p11.2p11.12(48,161,877–48,893,177)x4 | 11p11.2p11.12 | 732 | VUS | | Continue Pregnancy |
| 9 | Mild hydronephrosis、ventricular echogenic spots | arr[hg19] 11q22.3(105,992,927–106,726,576)x3 arr[hg19]22q11.21q11.22(21,974,284–23,193,408)x3 | 11q22.3 | 733 | VUS | Mother: arr[hg19] 11q22.3(105,992,927–106,726,576)x3, 22q11.21q11.22(21,974,284–23,190,569)x3; Father: arr(1–22)×2,(X,Y)×1 | Continue Pregnancy |
| | | | 22q11.21q11.22 | 1,219 | VUS | | |
| 10 | Mild hydronephrosis | arr[hg19] 14q12(32,791,611–33,057,809)x1 | 14q12 | 266 | VUS | | Continue Pregnancy |
| 11 | hydronephrosis | arr[hg19] 14q32.13q32.2(95,807,725–96,386,941)x3 | 14q32.13q32.2 | 579 | VUS | | Continue Pregnancy |
| 12 | hydronephrosis | arr[hg19] Yq11.223(24247140–25685346)x3 | Yq11.223 | 1,438 | VUS | | Continue Pregnancy |

Abbreviations: VUS = variants of uncertain significance.

gene, *ZNF92 (603974)*, which is an unexplored transcription factor with higher expression in human T lymphocytes [17] and breast cancer [18]. Privitera F et al. [19] found that carriers of *ZNF92 (603974)* variants were more likely to suffer from bipolar disorder. However, there is no report of an association between the 7q11.21 deletion and renal developmental abnormalities.Furthermore, one prenatal fetus with a 1q21.1 microduplication combined with a 14.2 microduplication was found. Approximately 10% of patients with 1q21.1 deletions have CAKUT, and the *PDZK1 (603831)* gene within this region is thought to be an important factor in urinary system abnormalities. Liao, C et al. [20] speculated that mutations in the *PDZK1 (603831)* gene might be significant in causing urinary system abnormalities. Levy, M et al. [21] found an association between 1q21.1 microduplication and neurodevelopmental abnormalities. Sanna-Cherchi, S et al. [22] discovered a high correlation between pathogenic CNVs in CAKUT and neuropsychiatric disorders. Bulu, E [23] reported a case of 17q12 deletion syndrome (17q12 deletion) associated with bipolar disorder, while Huang, Z [24] found that 17q12 deletion

**Table 3. Comparative analysis of the detection rates for fetal abnormalities in both groups with hydronephrosis via CMA.**

| Combine with other ultrasonic abnormalities | Cases (Cases) | Pathogenic abnormality | | VUS | | Subtotal | |
|---|---|---|---|---|---|---|---|
| | | Detected Number (Cases) | Detected Rate (%) | Detected Number (Cases) | Detected Rate (%) | Detected Number (Cases) | Detected Rate (%) |
| Isolated hydronephrosis | 36 | 2 | 5.56% | 7 | 19.44% | 9 | 25.00% |
| nonisolated hydronephrosis | 47 | 6 | 12.77% | 4 | 8.51% | 10 | 21.28% |
| Subtotal | 83 | 8 | 9.64% | 11 | 13.25% | 19 | 22.89% |
| $\chi^2$ | | 0.53 | | 1.28 | | 0.01 | |
| *p-Value* | | 0.4666 | | 0.2579 | | 0.9203 | |

**Table 4. Comparison of CMA identified fetal anomalies in severe and moderate hydronephrosis groups to the mild group.**

| Level | Cases (Cases) | Pathogenic abnormality | | VUS | | Subtotal | |
|---|---|---|---|---|---|---|---|
| | | Detected Number (Cases) | Detected Rate (%) | Detected Number (Cases) | Detected Rate (%) | Detected Number (Cases) | Detected Rate (%) |
| mild hydronephrosis | 40 | 1 | 2.5 | 3 | 7.5% | 4 | 10% |
| moderate to severe hydronephrosis | 43 | 7 | 16.28% | 9 | 20.93% | 16 | 37.21% |
| Subtotal | 83 | 8 | 9.64% | 12 | 14.46% | 20 | 24.10% |
| $\chi^2$ | | 3.07 | | 2.03 | | 6.97 | |
| *p-Value* | | 0.0797 | | 0.1542 | | 0.0083 | |

was the most common CNV in CAKUT patients, manifesting as congenital hydronephrosis and polycystic kidneys. Hydronephrosis may induce renal dysfunction, and renal abnormalities have been shown in numerous studies [25–29] to trigger neurological disorders. Given the link between CAKUT and neurodevelopmental abnormalities, it can be hypothesized that these CNVs of uncertain clinical significance might be the cause of hydronephrosis, although this requires further clinical data and functional research for validation.In summary, based on the results of the current retrospective study, CMA testing, when compared with karyotype analysis, did not significantly increase the detection rate of pathogenic variants. Therefore, it is not advisable to abandon karyotype analysis. However, due to its higher resolution, CMA testing can detect newly - discovered chromosomal copy number variations (CNVs). Although these newly detected CNVs bring new challenges to clinical diagnosis, they may be transformed into information with significant diagnostic value once verified through extensive clinical data and in-depth functional studies

This study used CMA to compare the incidence of chromosomal abnormalities between fetuses with isolated renal hydronephrosis and those with non-isolated renal hydronephrosis. The results were not statistically significant ($P > 0.05$), although this could be due to the small sample size. However, the detection rate of fetuses with mutations in the mild hydronephrosis group was statistically significantly different from that in the moderate to severe hydronephrosis group ($P < 0.05$). Therefore, the author still believes that as the grading level of renal hydronephrosis increases, CMA holds greater diagnostic value.

Consolidated, hydronephrosis associates with chromosomal abnormalities. The incidence of chromosome anomalies increases as the grade system advancing for hydronephrosis is elevated. Combinating diverse prenatal techniques can illuminate defects caused by chromosome microdeletions and/or microduplications, presenting significant value to clinical antenatal counseling.

## Author contributions

**Data curation:** Xiayuan Xu, Yue Qian, Liping Zhang, Jianfeng Luo.

**Funding acquisition:** Keqin Jin, Min Hu, Jianfeng Luo.

**Project administration:** Keqin Jin, Min Hu, Jianfeng Luo.

**Writing – original draft:** Keqin Jin, Min Hu.

**Writing – review & editing:** Keqin Jin.

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
