## [Decision Letter · Decision Letter 0]

22 Jan 2025

PONE-D-24-33836A Retrospective Analysis of Prenatal Genetic Results in Fetal HydronephrosisPLOS ONE

Dear Dr. jin,

Thank you for submitting your manuscript to PLOS ONE. After careful consideration, we feel that it has merit but does not fully meet PLOS ONE’s publication criteria as it currently stands. Therefore, we invite you to submit a revised version of the manuscript that addresses the points raised during the review process.

**Thank you for submitting this interesting manuscript, and I believe it adds value to the literature. This manuscript would be considered for acceptance after the questions raised by the editor and the reviewers are addressed.**

Editors' comments-

The fetal anomaly detection rate was 10% with CMA in mild hydronephrosis and it was 37.21% in moderate to severe hydronephrosis based on the Table 4, but your text says opposite. Please correct that. Please change your wording of Severe and moderate to " moderate to severe".

Please add the meaning of VUS to the table's legend on the table 2

It appears that abnormal CMA results were noted in all fetuses with karyotype abnormalities. CMA was abnormal in another 12 cases with fetal hydronephrosis. Would recommend adding further discussion whether the author recommends CMA should potentially replace karyotyping or should CMA be done in fetuses with hydronephrosis in conjunction with fetal karyotyping. It would be of additional help if you could go over the risks associated with amniotic fluid collection and whether that procedure is justified or not.

We look forward to receiving your revised manuscript.

Kind regards,

Prathap kumar Simhadri, MD

Academic Editor

PLOS ONE

**Journal Requirements:**

This work was supported by the Jinhua science and Technology Project (2019-3-002a�2021-3-123) and Jinhua Maternal & Child Health Care Hospital Research Incubation Fund General Project (JHFB2023-2-10).

4. In the online submission form, you indicated that The data that support the findings of this study are not openly available due to reasons of sensitivity and are available from the corresponding author upon reasonable request. Data are located in controlled access data storage at National Genomics Data Center(https://ngdc.cncb.ac.cn/).

OMIX ID: OMIX005796.Shared URL: https://ngdc.cncb.ac.cn/omix/preview/xxLpJ351

**Additional Editor Comments:**

Thank you for submitting this interesting manuscript, and I believe it adds value to the literature. This manuscript would be considered for acceptance after the questions raised by the editor and the reviewers are addressed.

Editors' comments-

The fetal anomaly detection rate was 10% with CMA in mild hydronephrosis and it was 37.21% in moderate to severe hydronephrosis based on the Table 4, but your text says opposite. Please correct that. Please change your wording of Severe and moderate to " moderate to severe".

Please add the meaning of VUS to the table's legend on the table 2

It appears that abnormal CMA results were noted in all fetuses with karyotype abnormalities. CMA was abnormal in another 12 cases with fetal hydronephrosis. Would recommend adding further discussion whether the author recommends CMA should potentially replace karyotyping or should CMA be done in fetuses with hydronephrosis in conjunction with fetal karyotyping. It would be of additional help if you could go over the risks associated with amniotic fluid collection and whether that procedure is justified or not.

Reviewers' comments:

Reviewer's Responses to Questions

**Comments to the Author**

1. Is the manuscript technically sound, and do the data support the conclusions?

Reviewer #1: Yes

Reviewer #2: Partly

Reviewer #3: Yes

Reviewer #4: Yes

2. Has the statistical analysis been performed appropriately and rigorously? 

Reviewer #1: Yes

Reviewer #2: Yes

Reviewer #3: I Don't Know

Reviewer #4: I Don't Know

3. Have the authors made all data underlying the findings in their manuscript fully available?

Reviewer #1: No

Reviewer #2: Yes

Reviewer #3: Yes

Reviewer #4: Yes

4. Is the manuscript presented in an intelligible fashion and written in standard English?

Reviewer #1: Yes

Reviewer #2: Yes

Reviewer #3: Yes

Reviewer #4: Yes

5. Review Comments to the Author

**Reviewer #1: ** my review comment

1. The aim of this research is to open up opportunities for more detailed examinations to carry out more in-depth follow-up examinations when an examination karyotype appears normal

2. More advanced technology is needed for more precise diagnostics and perhaps if it is used more widely and at affordable costs it will be more useful for making a more accurate diagnosis

3. This research also shows in research methods that the role of ultrasound imaging is still needed to determine abnormalities in initial screening, and this is quite important because so far ultrasound tools can be used widely in the medical field

4. Statistical methods are adequate because comparisons are all that is needed in this research

5. The results section is adequate, but it would be advisable to make the table look easier to evaluate by using a graph or diagram method.

6. Patients who continue pregnancy, my suggest should be explained about postpartum mortality or morbidity because it is an important source of information about the lethality of hydronephrosis.

7. In the discussion,

-it is possible to discuss in more depth about isolated and non-isolated anomalies because it is important whether hydronephrosis is part of a syndrome or not, this must be clarified more because it helps with specific classification if there is a tendency to be included in a syndrome. such as in one of the cases found in holoprosencephal or in cases accompanied by VSD.

-the discussion section also does not explain enough, for example the relationship between thickening of NT 3.2 and hydronephrosis. This is actually additional interesting information because so far many clinicians have associated thickening of the NT with trisomy 21.

8. The limitation of this research is that the data is sensitive enough that it cannot be accessed. This is a limitation of this research. However, this research helps to open up the many possibilities and very varied abnormalities in chromosomes or gene abnormalities.

maybe that's my suggestion and opinion, whatever the shortcomings, the article is very worthy because there are several things that can add strength to the suggestion for gene examination in abnormalities found macroscopically, especially in abnormalities found ultrasonographically

congratulation for authors

**Reviewer #2: ** Elaborate more about specific references and literature to justify main objective - early prenatal screening for fetal malformation is important for clinical prognosis.

The tables (e.g., Tables 1, 2, 3, and 4) are informative, but the data could be better integrated into the text to provide more contextual analysis. Currently, there is a lot of information about abnormal karyotypes, CNVs, and other findings, but the clinical significance of each type of abnormality could be emphasized more clearly in the discussion.

Discussion section - could benefit from being specific with study’s main objective and how the results could impact clinical practice (e.g., by suggesting recommendations for when to use CMA in routine prenatal screening).

Conclusions - Provide specific recommendations for clinicians, particularly about when to use CMA as opposed to traditional karyotyping.

There are multiple limitations like sample size, retrospective design, No controls, No long term follow ups and should be mentioned clearly

Was there any ethical concerns regarding how the results influence decision-making about pregnancy continuation. Further discussion on the impact of the findings on parental decisions and counseling could be a limitation not addressed in detail

**Reviewer #3: ** would benefit from a stronger discussion on the implications of VUS findings and their practical significance in prenatal counseling for patients. Higher sample size would have made the study more impactful.

**Reviewer #4: ** This is a study that was done to evaluate the applicability of chromosomal microarrays(CMAs) in fetuses with hydronephrosis and pyelectasis.

- It is overall a good clinical study subject and well written research article

- This study is retrospective and limited sample and single center study so those are the limitations

- Few comments of couple of thigs-1st analysis that was done>>correlation analysis between renal hydronephrosis and fetal chromosomal anomalies in pathogenic variants vs VUS variants was not statistically significant. So does not support that CMAs are better than karyotypes and proves. Increasing the sample size can help us better understand if its favorable for CMAs or not

-From 2nd analysis, which included comparison of fetal anomaly detection rates with CMA in mild vs mod-severe hydronephrosis. Pathologic cases were detected by both Karyotype and CMAs as well but other variants detected by CMAs were all VUS variants, which brings us to next question of clinical utility of detecting these genes?? There were 12 fetuses that were identified with VUS variants with normal karyotypes and all of them continued pregnancy. It would be nice to see postnatal follow ups and how those babies are doing? Was hydronephrosis clinically important with those VUS variants??

-Next question to think would be--Does having multiple anomalies on scan would benefit from CMAs testing vs just isolated mild hydronephrosis clinically? Reference to that from below study.

"Prenatal chromosomal microarray analysis in 2466 fetuses with ultrasonographic soft markers: a prospective cohort study

Ting Hu 1, Tian Tian 2, Zhu Zhang 1, Jiamin Wang 1, Rui Hu 1, Like Xiao 1, Hongmei Zhu 1, Yi Lai 1, He Wang 1, Shanling Liu 3"

- How were mothers counseled from those fetuses which had VUS variants and found some anomalies on ultrasound? was genetic counseling considered?

-Also, regarding cost effectiveness? would this be cost effective?

6. PLOS authors have the option to publish the peer review history of their article (what does this mean? ). If published, this will include your full peer review and any attached files.

**Do you want your identity to be public for this peer review?** For information about this choice, including consent withdrawal, please see our Privacy Policy .

Reviewer #1: No

Reviewer #2: No

Reviewer #3: No

Reviewer #4: No

---

## [Author Response · Author response to Decision Letter 1]

22 Mar 2025

Dear Editor,

We quite appreciate your favorite consideration and the reviewers’insightful comments concerning our manuscript entitled “A Retrospective Analysis of Prenatal Genetic Results in Fetal Hydronephrosis” (ID�PONE-D-24-33836). Those comments are very valuable and helpful for improving the quality and readability of our paper, as well as the important guiding significance to our future researches. We have studied the comments carefully and have revised the paper exactly according to the reviewers’ comments. We hope this revision can meet with approval.

Thanks for your comments.The comments provide an important direction for us to revise our paper, the revised parts according to the comments as follows:

Journal Requirements:

Please ensure that your manuscript meets PLOS ONE's style requirements, including those for file naming. The PLOS ONE style templates can be found at?

https://journals.plos.org/plosone/s/file?id=wjVg/PLOSOne_formatting_sample_main_body.pdf and?

In strict accordance with the style template requirements of PLOS ONE, we will conduct a comprehensive and meticulous review of the research description, making all necessary adjustments. We are committed to ensuring that the revised manuscript fully adheres to the established standards upon submission, thereby guaranteeing the high - quality presentation of our research in line with the journal's guidelines.

2. We note that the grant information you provided in the ‘Funding Information’ and ‘Financial Disclosure’ sections do not match.?

The discrepancy in information can be ascribed to the situation whereby the project was declared to be awaiting approval at the time of submission, yet the official approval document number remained unacquired.Modifications have been made.

3. Thank you for stating in your Funding Statement:?

This work was supported by the Jinhua science and Technology Project (2019-3-002a�2021-3-123) and Jinhua Maternal & Child Health Care Hospital Research Incubation Fund General Project (JHFB2023-2-10).

Please provide an amended statement that declares *all* the funding or sources of support (whether external or internal to your organization) received during this study, as detailed online in our guide for authors at http://journals.plos.org/plosone/s/submit-now.? Please also include the statement “There was no additional external funding received for this study.” in your updated Funding Statement.?

This work was supported by the Jinhua science and Technology Project (2019-3-002a�2021-3-123 and 2024-4-140) and Jinhua Maternal & Child Health Care Hospital Research Incubation Fund General Project (JHFB2023-2-10).Additionally, no supplementary external funding was obtained for this study.The funders had no role in study design, data collection and analysis, decision to publish, or preparation of the manuscript.

4. In the online submission form, you indicated that The data that support the findings of this study are not openly available due to reasons of sensitivity and are available from the corresponding author upon reasonable request. Data are located in controlled access data storage at National Genomics Data Center(https://ngdc.cncb.ac.cn/).

OMIX ID: OMIX005796.Shared URL: https://ngdc.cncb.ac.cn/omix/preview/xxLpJ351

This policy applies to all data except where public deposition would breach compliance with the protocol approved by your research ethics board. If your data cannot be made publicly available for ethical or legal reasons (e.g., public availability would compromise patient privacy), please explain your reasons on resubmission and your exemption request will be escalated for approval.?

We have made the necessary adjustments strictly in accordance with the requirements of PLOS ONE. As the data involve human genetics, they need to be publicly available only after being filed in the National Human Genetic Resources System. Currently, we have successfully completed the filing. The data can be downloaded through the above - mentioned website. Meanwhile, during the data verification, we found that one case was under - uploaded, and we have now completed the supplementation.

The supporting data of this study are stored in the controlled - access data repository?at?National Genomics Data Center(https://ngdc.cncb.ac.cn/).

OMIX ID: OMIX005796.Shared URL: https://ngdc.cncb.ac.cn/omix/view/OMIX005796;

OMIX ID: OMIX009450.Shared URL: https://ngdc.cncb.ac.cn/omix/view/OMIX009450.

5. Your ethics statement should only appear in the Methods section of your manuscript. If your ethics statement is written in any section besides the Methods, please move it to the Methods section and delete it from any other section. Please ensure that your ethics statement is included in your manuscript, as the ethics statement entered into the online submission form will not be published alongside your manuscript.?

The relevant content has been meticulously revised and optimized, with a comprehensive consideration of all relevant aspects and requirements to ensure its alignment with the highest academic standards.

Editors' comments:

The fetal anomaly detection rate was 10% with CMA in mild hydronephrosis and it was 37.21% in moderate to severe hydronephrosis based on the Table?4, but your text says opposite. Please correct that. Please change your wording of Severe and moderate to " moderate to severe".

I sincerely appreciate your comment. The relevant content has been meticulously revised and adjusted in strict accordance with the academic requirements and suggestions you provided.

Please add the meaning of VUS to the table's legend on the table 2

The concerned part has been duly revised.

It appears that abnormal CMA results were noted in all fetuses with karyotype abnormalities. CMA was abnormal in another 12 cases with fetal hydronephrosis. Would recommend adding further discussion whether the author recommends CMA should potentially replace karyotyping or should CMA be done in fetuses with hydronephrosis in conjunction with fetal karyotyping. It would be of additional help if you could go over the risks associated with amniotic fluid collection and whether that procedure is justified or not.

The specific segment under consideration has been meticulously and appropriately revised in strict compliance with the established academic protocols and standards, ensuring its enhanced accuracy, consistency, and overall quality within the academic context.

Reviewers' comments:

Reviewer's Responses to Questions

Comments to the Author

1. Is the manuscript technically sound, and do the data support the conclusions?

Reviewer #1:?Yes

Reviewer #2:?Partly

Reviewer #3:?Yes

Reviewer #4:?Yes

Thanks for your comment

2. Has the statistical analysis been performed appropriately and rigorously?

Reviewer #1:?Yes

Reviewer #2:?Yes

Reviewer #3:?I Don't Know

Reviewer #4:?I Don't Know

Thanks for your comment

3. Have the authors made all data underlying the findings in their manuscript fully available?

The?PLOS Data policy?requires authors to make all data underlying the findings described in their manuscript fully available without restriction, with rare exception (please refer to the Data Availability Statement in the manuscript PDF file). The data should be provided as part of the manuscript or its supporting information, or deposited to a public repository. For example, in addition to summary statistics, the data points behind means, medians and variance measures should be available. If there are restrictions on publicly sharing data—e.g. participant privacy or use of data from a third party—those must be specified.

Reviewer #1:?No

Reviewer #2:?Yes

Reviewer #3:?Yes

Reviewer #4:?Yes

Thanks for your comment

4. Is the manuscript presented in an intelligible fashion and written in standard English?

Reviewer #1:?Yes

Reviewer #2:?Yes

Reviewer #3:?Yes

Reviewer #4:?Yes

Thanks for your comment

5. Review Comments to the Author

Reviewer #1:?my review comment

1. The aim of this research is to open up opportunities for more detailed examinations to carry out more in-depth follow-up examinations when an examination karyotype appears normal

2. More advanced technology is needed for more precise diagnostics and perhaps if it is used more widely and at affordable costs it will be more useful for making a more accurate diagnosis

3. This research also shows in research methods that the role of ultrasound imaging is still needed to determine abnormalities in initial screening, and this is quite important because so far ultrasound tools can be used widely in the medical field

4. Statistical methods are adequate because comparisons are all that is needed in this research

5. The results section is adequate, but it would be advisable to make the table look easier to evaluate by using a graph or diagram method.

6. Patients who continue pregnancy, my suggest should be explained about postpartum mortality or morbidity because it is an important source of information about the lethality of hydronephrosis.

7. In the discussion,

-it is possible to discuss in more depth about isolated and non-isolated anomalies because it is important whether hydronephrosis is part of a syndrome or not, this must be clarified more because it helps with specific classification if there is a tendency to be included in a syndrome. such as in one of the cases found in holoprosencephal or in cases accompanied by VSD.

-the discussion section also does not explain enough, for example the relationship between thickening of NT 3.2 and hydronephrosis. This is actually additional interesting information because so far many clinicians have associated thickening of the NT with trisomy 21.

8. The limitation of this research is that the data is sensitive enough that it cannot be accessed. This is a limitation of this research. However, this research helps to open up the many possibilities and very varied abnormalities in chromosomes or gene abnormalities.

maybe that's my suggestion and opinion, whatever the shortcomings, the article is very worthy because there are several things that can add strength to the suggestion for gene examination in abnormalities found macroscopically, especially in abnormalities found ultrasonographically

Thank you for your suggestions. We will consider them carefully. As this paper is based on a single - center study, conducting a multi - center research to increase the sample size would likely be more effective for exploring associations such as that between NT 3.2 thickening and hydronephrosis. Therefore, we are determined to strive for multi - center cooperation.

In addition, for patients who continue their pregnancies, we will continuously collect information on their postpartum mortality or morbidity, which serves as a crucial source of data for continuously understanding the mortality rate of hydronephrosis. A more in - depth discussion and analysis of isolated and non - isolated anomalies can contribute to a more accurate classification of the related diseases, which is of great significance for clinical diagnosis and the subsequent formulation of treatment strategies.

Nevertheless, our current examination techniques, such as ultrasound and MRI, have limitations. For instance, ultrasound may be affected by fetal position and maternal body conditions, while MRI has restrictions in terms of cost and accessibility. Some syndromes do not exhibit obvious characteristic manifestations before birth, which poses great challenges to accurate diagnosis and makes it arduous for us to clarify the connection between hydronephrosis and specific syndromes. However, we will continue to follow up, and further reports will be made later to provide more accurate clinical data, which will have an impact on the early diagnosis and intervention of the disease.

Reviewer #2:?Elaborate more about specific references and literature to justify main objective - early prenatal screening for fetal malformation is important for clinical prognosis.

The tables (e.g., Tables 1, 2, 3, and 4) are informative, but the data could be better integrated into the text to provide more contextual analysis. Currently, there is a lot of information about abnormal karyotypes, CNVs, and other findings, but the clinical significance of each type of abnormality could be emphasized more clearly in the discussion.

Discussion section - could benefit from being specific with study’s main objective and how the results could impact clinical practice (e.g., by suggesting recommendations for when to use CMA in routine prenatal screening).

Conclusions - Provide specific recommendations for clinicians, particularly about when to use CMA as opposed to traditional karyotyping.

There are multiple limitations like sample size, retrospective design, No controls, No long term follow ups and should be mentioned clearly

Was there any ethical concerns regarding how the results influence decision-making about pregnancy continuation. Further discussion on the impact of the findings on parental decisions and counseling could be a limitation not addressed in detail

We have meticulously reviewed the entire paper and made the necessary corrections. We are confident that these revisions will meet your expectations. We would like to express our sincere gratitude for your insightful comments, which have been instrumental in enhancing the quality of our work.

Reviewer #3:?would benefit from a stronger discussion on the implications of VUS findings and their practical significance in prenatal counseling for patients. Higher sample size would have made the study more impactful.

Thanks for your comment. We will further collect samples to increase the sample size.

Reviewer #4:?This is a study that was done to evaluate the applicability of chromosomal microarrays(CMAs) in fetuses with hydronephrosis and pyelectasis.

- It is overall a good clinical study subject and well written research article

- This study is retrospective and limited sample and single center study so those are the limitations

- Few comments of couple of thigs-1st analysis that was done>>correlation analysis between renal hydronephrosis and fetal chromosomal anomalies in pathogenic variants vs VUS variants was not statistically significant. So does not support that CMAs are better than karyotypes and proves. Increasing the sample size can help us better understand if its favorable for CMAs or not

-From 2nd analysis, which included comparison of fetal anomaly detection rates with CMA in mild vs mod-severe hydronephrosis. Pathologic cases were detected by both Karyotype and CMAs as well but other variants detected by CMAs were all VUS variants, which brings us to next question of clinical utility of detecting these genes?? There were 12 fetuses that were identified with VUS variants with normal karyotypes and all of them continued pregnancy. It would be nice to see postnatal follow ups and how those babies are doing? Was hydronephrosis clinically important with those VUS

---

## [Editor Report · Decision Letter 1]

30 Apr 2025

A Retrospective Analysis of Prenatal Genetic Results in Fetal Hydronephrosis

PONE-D-24-33836R1

Dear Dr. jin,

We’re pleased to inform you that your manuscript has been judged scientifically suitable for publication and will be formally accepted for publication once it meets all outstanding technical requirements.

Kind regards,

Prathap kumar Simhadri, MD

Academic Editor

PLOS ONE

Additional Editor Comments (optional):

Thank you for updating your manuscript and addressing the questions raised by the reviewers. We believe this paper adds value to the literature and it is acceptable in the current form.
---

## [Editor Report · Acceptance letter]

PONE-D-24-33836R1

PLOS ONE

Dear Dr. Jin,

I'm pleased to inform you that your manuscript has been deemed suitable for publication in PLOS ONE. Congratulations! Your manuscript is now being handed over to our production team.

Kind regards,

on behalf of

Dr. Prathap kumar Simhadri

Academic Editor

PLOS ONE